# In Situ Grafting of Silica Nanoparticle Precursors with Covalently Attached Bioactive Agents to Form PVA-Based Materials for Sustainable Active Packaging

**DOI:** 10.3390/polym13172889

**Published:** 2021-08-27

**Authors:** Miri Klein, Anat Molad Filossof, Idan Ashur, Sefi Vernick, Michal Natan-Warhaftig, Victor Rodov, Ehud Banin, Elena Poverenov

**Affiliations:** 1Agro-Nanotechnology and Advanced Materials Center, The Department of Food Science, Agricultural Research Organization, The Volcani Center, Rishon LeZion 7505101, Israel; miri2222@gmail.com (M.K.); anatmolad@gmail.com (A.M.F.); 2Agricultural Engineering, Sensing, Information and Mechanization Engineering, Agricultural Research Organization, The Volcani Center, Rishon LeZion 7505101, Israel; idana@volcani.agri.gov.il (I.A.); sefi@volcani.agri.gov.il (S.V.); 3Faculty of Life Sciences, The Institute for Advanced Materials and Nanotechnology, Bar Ilan University, Ramat-Gan 5290002, Israel; natan.michal@gmail.com (M.N.-W.); Ehud.Banin@biu.ac.il (E.B.); 4Department of Postharvest Science of Fresh Produce, Agricultural Research Organization, The Volcani Center, Rishon LeZion 7505101, Israel; vrodov@volcani.agri.gov.il

**Keywords:** polyvinyl alcohol, silica nanoparticle, active package, antioxidant, antimicrobial

## Abstract

Sustainable antibacterial–antioxidant films were prepared using in situ graftings of silica nanoparticle (SNP) precursors with covalently attached bioactive agents benzoic acid (ba) or curcumin (cur) on polyvinyl alcohol (PVA). The modified PVA-SNP, PVA-SNP-ba and PVA-SNP-cur films were characterized using spectroscopic, physicochemical and microscopic methods. The prepared films showed excellent antibacterial and antioxidant activity, and increased hydrophobicity providing protection from undesired moisture. The PVA-SNP-ba films completely prevented the growth of the foodborne human pathogen Listeria innocua, whereas PVA-SNP-cur resulted in a 2.5 log reduction of this bacteria. The PVA-SNP-cur and PVA-SNP-ba films showed high antioxidant activity of 15.9 and 14.7 Mm/g TEAC, respectively. The described approach can serve as a generic platform for the formation of PVA-based packaging materials with tailor-made activity tuned by active substituents on silica precursors. Application of such biodegradable films bearing safe bioactive agents can be particularly valuable for advanced sustainable packaging materials in food and medicine.

## 1. Introduction

Advanced packaging materials have recently become a subject of great scientific and applied interest [1,2,3]. Public tendencies and official regulations are encouraging the search for biodegradable alternatives to fossil fuel-sourced packaging materials, which cause huge environmental damage due to their massive use. In addition, modern packaging materials are expected to serve as more than passive wrapping, by protecting the encased product and enhancing its quality and storability. Materials capable of oxygen scavenging, moisture absorption, carbon dioxide control, antimicrobial protection, ethylene absorption and neutralization of off-flavors are in high demand [1,2,3,4].

Microbial contamination and oxidation are the biggest problems in the field of food, pharmacology, medicine and cosmetics. Oxidative processes can cause qualitative damage, affecting the color, flavor and structure of the packed products, and can convert their beneficial active components to inefficient or even harmful compounds [5,6,7]. The most common solution to these problems is the addition of oxygen scavengers, metal oxides, melanin and nature-sourced antioxidants to the packaging materials [8,9]. Harmful microorganisms cause product spoilage and can be agents of serious diseases in the case of human pathogens. Numerous approaches for the design of antimicrobial packaging materials have been reported, among them the use of antimicrobial copolymers as raw material, incorporation of antimicrobials during the extrusion process, covalent linkage of bactericide moieties, use of superhydrophobic materials to repel bacteria and more [10,11,12,13,14,15]. Utilizing nature-sourced bioactive materials such as phosvitin with gallic acid [16] or exopolysaccharides of *L. actobacillus plantarum* [17] were reported to extend food shelf life.

Surface grafting of nanoparticles is an effective and feasible method for forming active packaging materials. Metal nanoparticles such as silver, gold, copper, etc., have been variously grafted on polymer surfaces, conferring antimicrobial properties to the resultant materials [18,19,20]. Cyclodextrin nanoparticles have also been grafted on polymer surfaces and used as carriers for antifungal and antibacterial preserving agents [21]. Sonication was reported to successfully functionalize polymers [3], for instance, carbon nanotube–vitamin B1 composites added to polyvinyl alcohol (PVA) via sonication [22]. Multilayer assembly of polyelectrolytes is another approach to introducing a functional coating on a polymer surface, where the added layers can be used to carry active antioxidant agents [23]. The sol–gel method is also used for surface grafting. In this regard, silica nanoparticles (SNPs) have superior potential due to their excellent biocompatibility and stability. Bare SNPs grafted on polyethylene, polystyrene, poly(ethylene) glycol and polyvinyl chloride polymers were reported to reduce bacterial adhesion by increasing surface roughness [24]. In addition, SNPs have been used for filling or co-condensing with hindered phenols, providing the grafted polymers with antioxidant properties [25].

Preparation of edible packaging materials which are biodegradable and produced from agro-industrial by-products or waste have important economic and environmental benefits [26,27,28]. For instance, González et al. prepared soy protein films reinforced with cellulose nanofibers obtained from soybean hulls and pods. The prepared films have demonstrated homogeneous surfaces and good mechanical properties [27]. Wang et al. developed edible materials based on whey protein isolate nanofibers with glycerol as a plasticizer and carvacrol as an antimicrobial agent. The resulting edible films exhibited high antibacterial activity and weight loss decrease in salted duck egg yolk [28].

In this work, we aimed to form active packaging materials that consist entirely of sustainable components. Curcumin and benzoic acid-substituted silica precursors were, for the first time, synthesized and grafted as soft nanoparticles on polyvinyl alcohol (PVA), a biocompatible polymer that is widely used as a packaging material and food supplement and for other applications [12,29,30,31]. Curcumin is a natural polyphenolic that has a broad activity spectrum, including antibacterial, antiviral, antifungal and antioxidant properties [32]. Benzoic acid is a natural (but can also be synthetic) aromatic compound that is also widely used as a safe and effective antibacterial agent against human pathogens [33]. In this work, the bare SNPs and the modified precursor-based SNP-curcumin (SNP-cur) and SNP-benzoic acid (SNP-ba) were grafted in situ on PVA by the sol–gel method. The prepared PVA-SNP, PVA-SNP-cur and PVA-SNP-ba films were comprehensively characterized and their antibacterial and antioxidant activities were studied.

The physical and chemical properties of the prepared PVA-SNP, PVA-SNP-cur and PVA-SNP-ba films were studied by the following methods: Nuclear magnetic resonance (NMR) and attenuated total reflectance-Fourier transform infrared (ATR-FTIR) spectroscopy were used to confirm the synthesis of SNP-cur and SNP-ba precursors and their success of grafting on PVA films. Thermal stability and degradation of the prepared materials were studied by thermogravimetric analysis (TGA) as well as differential scanning calorimetry (DSC). In addition, water contact angle, thickness and water vapor permeability were measured to understand the effect of the modification on films’ hydrophobicity. High-resolution scanning electron microscopy (HR-SEM) and atomic force microscopy (AFM) were used to study the morphology of the surfaces of the film. 

## 2. Materials and Methods

### 2.1. Materials

The following analytical-grade chemicals were purchased from Sigma-Aldrich (Israel) and were used as received: polyvinyl alcohol (PVA) white to off-white powder (Mw 89,000–98,000, 99+% hydrolyzed), curcumin, benzoic acid, 1,1′-carbonyldiimidazole, tetraethyl orthosilicate (TEOS) (99.999% purity), Trolox [(±)-6-hydroxy-2,5,7,8-tetramethylchromane-2-carboxylic acid] (97% purity) and 28% (*v*/*v*) aqueous NH_4_OH (99.99% purity). (3-Isocyanatopropyl)triethoxy silane, triethylamine, and (3-aminopropyl)-triethoxysilane were purchased from Alfa Aesar (Haverhill, MA, USA). Anhydrous tetrahydrofuran (THF) and anhydrous dichloromethane (DCM) were purchased from Acros Organics (Geel, Belgium). Acetone, n-hexane and ethanol absolute (EtOH) of analytical reagent (AR) grade were purchased from Gadot (Israel) and used without further purification.

### 2.2. Synthesis

#### 2.2.1. PVA-SNP-cur

First, curcumin-based silicate was synthesized: (3-isocyanatopropyl)triethoxy silane (81 µL, 0.326 mmol) and triethylamine (38 µL, 0.271 mmol) were added simultaneously to a mixture of curcumin (100 mg, 0.271 mmol) dissolved in dry acetone (10 mL) under nitrogen atmosphere. The reaction mixture was stirred at room temperature for 4 h until no reactant was detectable by thin-layer chromatography (TLC). The solvent was evaporated and hexane was added to precipitate the product. The residue was washed several times with hexane to remove triethylamine and excess isocyanate, giving the curcumin-based silicate (206 mg, 0.2387 mmol, 88%). ^1^H-NMR measurements were performed verifying the expected chemical structure and purity of the prepared materials. Each chemical shift was related to the appropriate H atom in the prepared precursor molecule. The peak’s split and integration was in accordance with the expected structure.

^1^H-NMR (400 MHz, CDCl_3_): δ 0.69 (4H, t, J = 8.1 Hz), 1.24 (18H, t, J = 7 Hz), 1.71 (4H, quintet, J = 8 Hz), 3.28 (4H, q, J = 6.4 Hz), 3.84 (12H, q, J = 7 Hz), 3.89 (6H, s), 5.38 (2H, t, J = 5.9 Hz), 5.85 (2H, s), 6.5 (2H, d, J = 15.8 Hz), 7.11–7.16 (6H, m), 7.62 (2H, d, J = 15.8 Hz).

Then, the prepared curcumin-silicate precursor was grafted in situ on PVA film to form a PVA film with grafted silica-curcumin nanoparticles (PVA-SNP-cur): PVA films (5 × 2 cm^2^) were precleaned in an ultrasonic bath (Elmasonic S 30 ultrasonic bath, 37 kHz) at full-power irradiation for 15 min in EtOH. Then they were placed in a reaction vessel containing 15 mL EtOH. The reactants were added to the vessel in the following order: 2.5 mL (11.2 mmol) TEOS and 400 mg (0.463 mmol) curcumin-silicate precursor, pre-dissolved in a minimal amount of acetone, and 7.5 mL distilled water. The reaction mixture was stirred for 30 min at 60 °C, and then 3 mL of aqueous NH_4_OH (28% *w*/*w* in H_2_O) was added. The reaction was run at room temperature for 2 days under constant agitation (orbital shaker). The resulting modified films were washed with EtOH (×3) for 10 min using an ultrasonic bath to remove physically adsorbed SNP-cur. The obtained PVA-SNP-cur films were then left to air dry before characterization and further processing.

#### 2.2.2. PVA-SNP-ba

First, benzoic acid silicate was synthesized: 1,1′-carbonyldiimidazole (4 g, 24.6 mmol) was added to benzoic acid (3 g, 24.6 mmol) dissolved in a mixture of dry dichloromethane (30 mL) and dry tetrahydrofuran (30 mL), and the reaction mixture was stirred at room temperature for 1 h. Next, (3-aminopropyl)-triethoxysilane (APTES; 6.9 mL, 29.5 mmol) was added, and the reaction mixture was stirred at room temperature for an additional 18 h. When the reaction was completed (as determined by TLC), the medium was concentrated under vacuum and purified by column chromatography on silica gel (SiliaFlash Irregular Silica Gel G60, 60–200 µm, 60 Å, 70–230 mesh) (hexane:ethyl acetate = 70:30 *v/v*) to achieve the pure oily colorless benzoic acid-based silicate (6.64 g, 20.4 mmol, 83%). ^1^H-NMR measurements were performed verifying the expected chemical structure and purity of the prepared materials. Each chemical shift was related to the appropriate H atom in the prepared precursor molecule. The peak’s split and integration were in accordance with the expected structure. ^1^H-NMR (400 MHz, CDCl_3_): δ 0.69 (2H, t, J = 8 Hz), 1.2 (9H, t, J = 7 Hz), 1.74 (2H, quintet, J = 7.9 Hz), 3.44 (2H, q, J = 6.5 Hz), 3.81 (6H, q, J = 7 Hz), 6.62 (1H, br s), 7.39 (2H, t, J = 7.2 Hz), 7.46 (1H, t, J = 7.4 Hz), 7.76 (2H, d, J = 7.2 Hz).

Then, the prepared benzoic acid-silicate precursor was grafted in situ on PVA films to form PVA with grafted silica-benzoic acid nanoparticles (PVA-SNP-ba): PVA films (5 × 2 cm^2^) were precleaned as in Section 2.2.1. Then they were placed in a reaction vessel containing 15 mL of EtOH, and the reactants were added in the following order: 2.5 mL (11.2 mmol) TEOS, 500 mg (1.538 mmol) benzoic acid-silicate precursor, and 7.5 mL distilled water. The reaction mixture was stirred for 30 min at room temperature, and then 3 mL of aqueous NH_4_OH (28% *w/w* in H_2_O) was added. The reaction was run at 50 °C for 2 days under constant agitation (orbital shaker). The resulting modified films were washed with EtOH (×3) for 10 min using an ultrasonic bath to remove physically adsorbed SNP-ba, and left to air dry before characterization and further processing.

#### 2.2.3. PVA-SNP

PVA films (5 × 2 cm^2^) were precleaned as in Section 2.2.1. Then they were placed in a reaction vessel containing 15 mL EtOH and 7.5 mL distilled water and 2.5 mL (11.2 mmol) TEOS was added. The reaction was stirred for 15 min at 50 °C, and then 3 mL of aqueous NH_4_OH (28% *w/w* in H_2_O) was added. The reaction was run at 50 °C for 2 days under constant agitation (orbital shaker). The resulting modified films were washed with EtOH (×3) for 10 min using an ultrasonic bath to remove physically adsorbed SNPs, then left to air dry before characterization and further processing.

### 2.3. Characterization

#### 2.3.1. Nuclear Magnetic Resonance (NMR) Spectroscopy

The NMR spectra were recorded on a Bruker AMX 400 MHz spectrometer (Bruker, United Kingdom). ^1^H-NMR signals are reported in ppm downfield from tetramethylsilane (TMS). ^1^H signals are referenced to the residual proton of a deuterated solvent, 7.26 ppm for CDCl_3_. All measurements were performed at 22 °C in CDCl_3_.

#### 2.3.2. Attenuated Total Reflectance-Fourier Transform Infrared (ATR-FTIR)

ATR-FTIR spectroscopy was performed using a Thermo Scientific Nicolet iS5 FTIR spectrometer (Thermo Scientific, Germany). The films were subjected to 32 scans at 4 cm^−1^ resolution between 500 and 4000 cm^−1^.

#### 2.3.3. Thermal Analyses

Thermal stability and degradation of all the polymer films were analyzed by thermogravimetric analysis (TGA) in a PerkinElmer TGA 8000 (PerkinElmer, Shelton, CT, USA), and differential scanning calorimetry (DSC) in a PerkinElmer DSC 6000 instrument (PerkinElmer, Shelton, CT, USA). TGA weight-loss curves were recorded under N_2_ (20 mL/min) at a heating rate of 10 °C/min from 50 °C to 800 °C for each polymer treatment. DSC thermograms were recorded under nitrogen at a heating rate of 5 °C/min over the temperature range of 25–250 °C.

#### 2.3.4. High-Resolution Scanning Electron Microscopy (HR-SEM)

HR-SEM analysis was performed using a field emission scanning electron microscope model MiraTescan (Tescan, Brno, Czech Republic) with an acceleration voltage of 7.0 kV and a secondary electron detector. Each sample was fixed onto a stub with a carbon adhesive tape and Pd/Au-sputtered (40 mA, 30 s) before analysis.

#### 2.3.5. Contact-Angle Measurements

The surface hydrophobicity of all PVA-based films was analyzed through contact-angle measurements using a goniometer KRÜSS DSA 100 model (KRÜSS, Hamburg (Germany). For each sample, an image of a water droplet placed on the sample surface was captured by a CCD camera and then analyzed using the DSA3 software: 4 μL of distilled water was introduced on the surface of each film sample and the static contact angle of the water droplet on the surface was measured. Measurements were performed in three different positions for each film and the average values are reported.

#### 2.3.6. Film Thickness

A micrometer (Mitutoyo, M820-25) with a resolution of 0.001 mm in the range of 0–25 mm was used to measure the thickness of the films. Each film was measured at seven random positions and the average thickness of each film is reported.

#### 2.3.7. Water Vapor Permeability (WVP)

The WVP of the films was measured as follows: conical tubes (15 mL) containing 5 mL distilled water were covered with the respective film samples and sealed to prevent leakage. These tubes were placed in a desiccator and maintained at room temperature and 75% relative humidity for 96 h to ensure complete saturation of the film samples. The weight difference of the tubes before and after 96 h was noted. The WVP of the film samples was calculated according to equation 1, where Δw is the weight difference of the tubes (g), l is the film thickness (m), A is the film area (m^2^), ΔP is the vapor pressure difference (3170 Pa at 25 °C) and t is the permeation time (s).
WVP = (Δw × l)/(A × ΔP × t)(1)

#### 2.3.8. Atomic Force Microscopy (AFM)

Topographic imaging was performed using an Innova AFM with a NanoDrive Controller (Bruker, Fremont, CA, USA) operating in tapping mode, in air, at room temperature. Surface images, using scan widths ranging from 1 µm to 5 µm with a scan rate of 1.0 Hz, were acquired at fixed resolution (512 × 512 data points). Bruker 0.01e0.025 Ohm-cm Antimoni (n) doped silicon tips (model: RTESPA-CP) were used. The roughness parameter was calculated for the scanned area (5 × 5 µm^2^) using NanoScope Analysis software. The AFM images and roughness calculations were obtained for different places on the sample and the most typical areas are presented.

### 2.4. Inhibition of Bacterial Growth

A fresh colony of Listeria innocua ATCC 33090 was transferred to a square 0.5-cm^2^ film surface (PVA, PVA-SNP, PVA-SNP-cur and PVA-SNP-ba films) using a sterile bacteriological needle. The bacteria were evenly spread on the sample and incubated overnight at 25 °C. Following the incubation step, the cells were removed using a cell scraper and 200 µL of sterile water and then 800 µL of sterile water was added to complete to 1 mL. The scraped cells were then transferred together with the film into an Eppendorf tube containing 650 µL lithium broth (LB). The Eppendorf was vortexed vigorously for 1 min to release any surface-attached bacteria that did not come off during the scraping stage. The bacteria were serially diluted and plated on LB agar plates. Following overnight incubation, the number of colony forming units (CFU) was determined. All experiments were conducted in duplicates at least three independent times.

### 2.5. Antioxidant Activity

First, the reagent ABTS+·(azinobis-3-ethylbenzothiazoline-6-sulfonate) and 1.0 mM Trolox standard were prepared. Reagent ABTS+ radicals were prepared in acetate buffer (50 mM pH 4.3) as described in Vinokur et al. [34] but with 59 µM potassium persulfate (K_2_S_2_O_8_) as the radical initiator. The reagent ABTS+·is a clear solution of greenish-blue color, with optical density (A_734_) ranging from 0.5 to 0.7. The 1.0 mM Trolox standard was prepared by dissolving 2.5 mg of Trolox in 100 µL ethanol and diluting to 10 mL with acetate buffer (50 mM, pH 4.3). The film samples (PVA, PVA-SNP, PVA-SNP-cur and PVA-SNP-ba) were then cut into smaller pieces and placed on four different plastic cuvettes containing 2 mL ABTS+·solution. Each experimental film was weighed. In addition, standard and blank samples were prepared by addition of 10 µL Trolox standard or 10 µL buffer to ABTS+·solution tubes, respectively. The tubes were incubated at room temperature for 15 min with vortexing every 3–5 min. The absorption in wavelength A_734_ of the different samples was studied in a Jenway 6505 UV–Vis spectrophotometer (Shimadzu, Japan). Acetate buffer was used as the reference.

The antioxidant activity of the tested films in Trolox equivalents (TE, mM) was calculated according to Equation (2):TE = Cstandard · (Ablank film − Asample film)/(Ablank − Astandard)(2)
where Cstandard is the standard concentration, and Asample film, Ablank film, Ablank and Astandard are absorbance values of the sample film, blank film, blank solution (acetate buffer) and standard solution (Trolox), respectively.

Trolox equivalent antioxidant capacity (TEAC) per film weight was calculated according to Equation (3):TEAC (mM TE/g) = (TE·V)/W(3)
where V is the ABTS+· reagent volume in a test tube (in mL), and W is the weight of the film sample (in g). For calculations, the assay was performed in three replicates.

## 3. Results and Discussion

### 3.1. Synthesis, Characterization and Physicochemical Properties

The silica precursors with covalently attached bioactive agents curcumin and benzoic acid were synthesized and grafted in situ as nanoparticles on PVA by sol–gel method to form PVA-SNP-cur and PVA-SNP-ba films, respectively. PVA-SNP films that contained bare SNPs were also formed with TEOS using the sol–gel method (Figure 1). All the prepared films were colorless and transparent except the PVA-SNP-cur which was yellowish.

ATR-FTIR, TGA and DSC studies were performed. The FTIR spectra confirmed the success of the grafting process (Figure 2). The original PVA film demonstrated characteristic peaks at 3550 and 3200 cm^−1^ assigned to O–H stretching due to the intermolecular and intramolecular hydrogen bonds of the PVA polymer. The vibrational band peak observed between 2800 and 3000 cm^−1^ referred to C–H stretching from the alkyl groups. Upon grafting of bare SNPs, a new peak in the 1000–1130 cm^−1^ region was observed, indicating the formation of Si–O bonds. This peak was also observed in the PVA-SNP-cur and PVA-SNP-ba films. In addition, these two latter films exhibited characteristic carbonyl bands at 1520–1600 cm^−1^.

In the TGA (Figure 3A), all studied films showed first weight loss of ~7.5% related to moisture evaporation at ~60 °C. The difference in thermal decomposition behavior between the original PVA and the modified PVA films can be clearly seen from the calculated onset of the second decomposition region (362 °C for all SNP-grafted PVA films vs. 262 °C for the original PVA). On the other hand, the thermal decomposition profiles did not differ much between the PVAs grafted with different SNP types. The increased temperature for the second decomposition onset indicated enhanced thermal stability of the film upon SNP grafting. The final decomposition of the original PVA film with a weight loss of ~100 wt% was observed at 650 °C, whereas the grafted films decomposed at 450 °C with a weight loss of 89–93 wt%.

DSC studies of the original and grafted PVA films were performed, and spectra of the first and second heating runs are presented (Figure 3B). The first heating run gave a single broad endothermic peak at 100–107 °C attributed to the moisture loss. Because PVA tends to absorb water, to eliminate this effect, a second heating run was performed. On the second-run curve, the endothermic peak at 100 °C disappeared, confirming its attribution to water, and a new exothermic peak, attributed to PVA degradation, was detected above 225 °C. There were no significant differences in the exothermic PVA degradation temperature among the PVA films, indicating that SNP grafting does not cause significant changes in the film’s internal structure. The glass transition temperature (Tg) obtained for the PVA films was ~76 °C, close to that previously reported in the literature [35].

The studied films’ morphology was examined by HR-SEM (Figure 4). The original PVA film demonstrated a smooth surface. The SNP-, SNP-cur- and SNP-ba-grafted films showed spherical particles of 50–120 nm size. Whereas in all film types, the nanoparticles were quite homogeneously distributed, SNP-cur demonstrated a particularly dense and consistent cover.

Topographic imaging of PVA, PVA-SNP, PVA-SNP-cur and PVA-SNP-ba films by atomic force microscopy (AFM) confirmed the appearance of spherical homogeneous nanoparticles on the PVA surface upon grafting. The grafted SNPs, SNP-cur and SNP-ba increased the overall roughness of the surface. Rq values increased from 4.6 nm for the original PVA film to 16.3, 42.9, and 34.3 nm for PVA-SNP, PVA-SNP-cur and PVA-SNP-ba surfaces, respectively (Figure 5).

The contact angle of a drop of water on the prepared films was measured (Table 1). The introduction of silica nanoparticles increased the hydrophobicity of the grafted films. Interestingly, even bare SNPs increased the surface hydrophobicity (87° for PVA-SNP vs. 32° for the original PVA), despite numerous hydroxyl groups on their surface. This effect can probably be explained by topographic changes since the grafted SNPs provided the PVA surface with higher roughness. PVA-SNP-ba films demonstrated the highest hydrophobicity (102°), while PVA-SNP-cur films were found to be significantly less hydrophobic (52°). Both ba and curcumin are relatively hydrophobic molecules that are expected to increase the hydrophobicity of the grafted films. At the molecular structure level, the low hydrophobicity of PVA-SNP-cur could be explained by the presence of numerous polar groups (carbonyl, hydroxyl and methoxy) on the curcumin molecule. Moreover, the curcumin molecule has a more resonance structure, compared to ba, at which the oxygen atom has a negative charge which makes it less hydrophobic than PVA-SNP-ba. In addition, as can be seen on the HR-SEM (Figure 4), SNP-cur provided a very uniform coverage of the PVA film, resulting in a smoother structure. Surface roughness and a material’s hydrophobicity are known to be closely related [36,37]. The impact of silica nanoparticles on surface hydrophobicity has been numerously reported [31]. For instance, Gao et al. [38] prepared highly hydrophobic (water contact angle of up to 155°) cotton and polyester fabrics using silica nanoparticles.

Hydrophobicity is an extremely important parameter for numerous applications. Used as a packaging material, the increased hydrophobicity of the modified polymer can provide additional protection from undesirable moisture and reduce the adhesion of harmful microorganisms [31,37].

Another parameter that is particularly important for materials intended for packaging is WVP, because less water transport increases the shelf life of the product. The WVP was reduced upon grafting of all SNP types on the PVA films, with PVA-SNP-ba demonstrating the lowest permeability to water (0.24 × 10^−10^ vs. 0.81 × 10^−10^ g s^−1^ m^−1^ Pa^−1^ for the original PVA) correlating with it having the highest hydrophobicity (Table 1). Interestingly, PVA-SNP films showed higher water vapor permeability than PVA-SNP-cur, despite the fact that PVA-SNP-cur has a smaller water contact angle in comparison to PVA-SNP. Probably, in the case of PVA-SNP, an increase in water contact angle is achieved due to the high surface roughness. However, unlike PVA-SNP-cur or PVA-SNP-ba, PVA-SNP films do not have grafted hydrophobic molecules. Therefore, their ability to inhibit the transport of water molecules is lower than in the case of ba or cur-grafted films.

### 3.2. Antimicrobial and Antioxidant Activity

The antimicrobial activity of the prepared films was examined against one of the most dangerous foodborne human pathogens, *Listeria innocua* [34]. The PVA-SNP-ba film demonstrated outstanding antimicrobial activity, resulting in complete prevention of bacterial growth; PVA-SNP-cur showed 2.5 log reduction and PVA grafted with bare SNPs a 1.0 log reduction in bacteria growth (Figure 6 left). The combination of surface roughness of the silica nanoparticles and the bactericidal properties of the attached active substituent therefore likely conferred antimicrobial properties of the prepared PVA-SNP-ba and PVA-SNP-cur. It can be seen that PVA-SNP-ba had higher antimicrobial activity against *Listeria innocua* than PVA-SNP-cur. An excellent antimicrobial activity of PVA-SNP-ba correlates with previously reported results, where benzoic acid was reported to have a strong ability to damage Listeria’s phospholipid membrane, perhaps because of its lipophilic features [39].

Antioxidant activity of the grafted PVA films was measured as Trolox equivalents [40]: the higher the Trolox equivalent antioxidant capacity (TEAC) in mM/g, the stronger the antioxidative activity of the materials (Figure 6 right). PVA films grafted with SNP-cur or SNP-ba exhibited much higher antioxidant activity than PVA grafted with bare SNPs (15.9, 14.7 and 6.5 TEAC mM/g, for PVA-SNP-cur, PVA-SNP-ba and PVA-SNP, respectively). The TEAC values were calculated using the original unmodified PVA films as a blank. The excellent antioxidant properties of the modified PVA-SNP-cur and PVA-SNP-ba could be very useful for their application as active packaging materials in protecting the quality of oxidation-sensitive products and extending their shelf life.

## 4. Conclusions

We described the preparation of antibacterial–antioxidant PVA films using in situ graftings of SNP precursors with covalently attached benzoic acid or curcumin as active agents. The prepared PVA-SNP-ba demonstrated total growth inhibition of the foodborne human pathogen *Listeria innocua*, and PVA-SNP-cur showed excellent antioxidant activity. SNP/SNP-cur/SNP-ba grafting was also found to increase the PVA film’s hydrophobicity, conferring to the modified materials an ability of protection from undesired moisture. This study suggests a feasible approach to forming sustainable PVA-based materials with tailor-made activity, which can be tuned by an active substituent on the silica precursor. Such biodegradable films, bearing safe bioactive agents, can be particularly valuable for packaging applications in food, medicine, pharmacology and cosmetics. This study’s findings can further serve as a platform for the formation of active materials for applications in other fields.

## Figures and Tables

**Figure 1 polymers-13-02889-f001:**
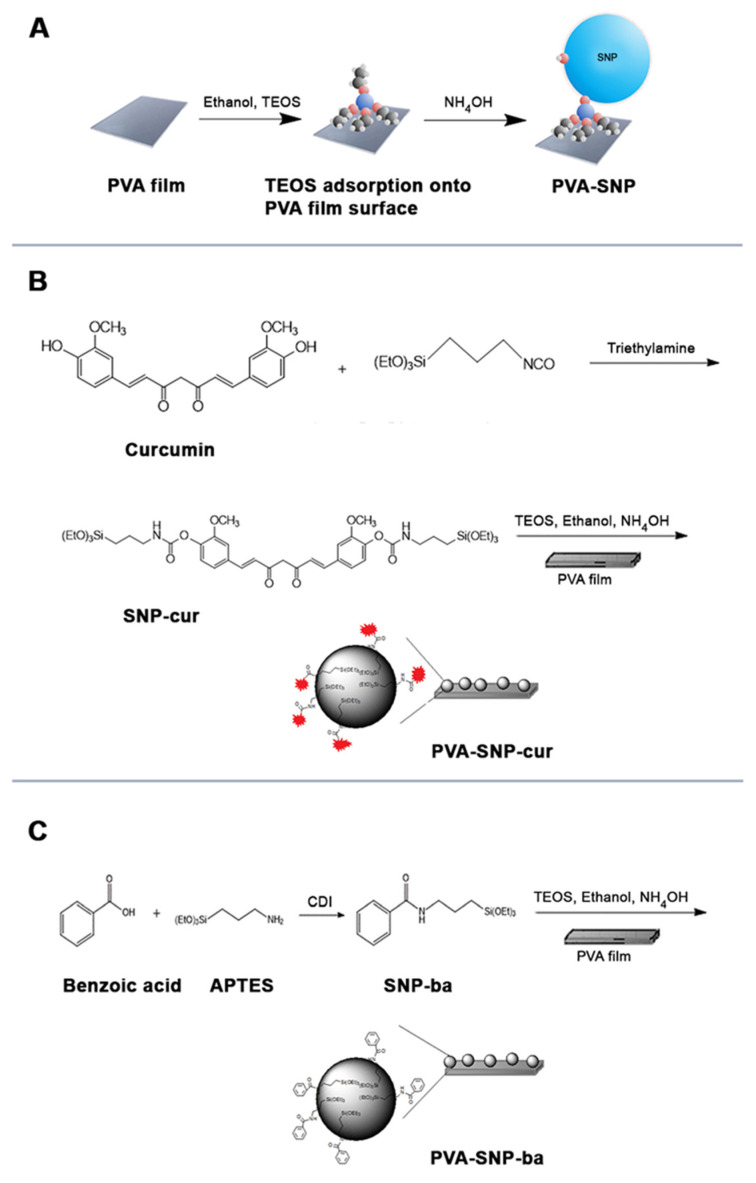
(**A**) In situ grafting of tetraethyl orthosilicate (TEOS) on PVA to form PVA-SNP films. (**B**) Preparation of curcumin-bearing silica precursor and in situ grafting on PVA to form PVA-SNP-cur films. (**C**) Preparation of benzoic acid-bearing silica precursors and in situ grafting on PVA to form PVA-SNP-ba films.

**Figure 2 polymers-13-02889-f002:**
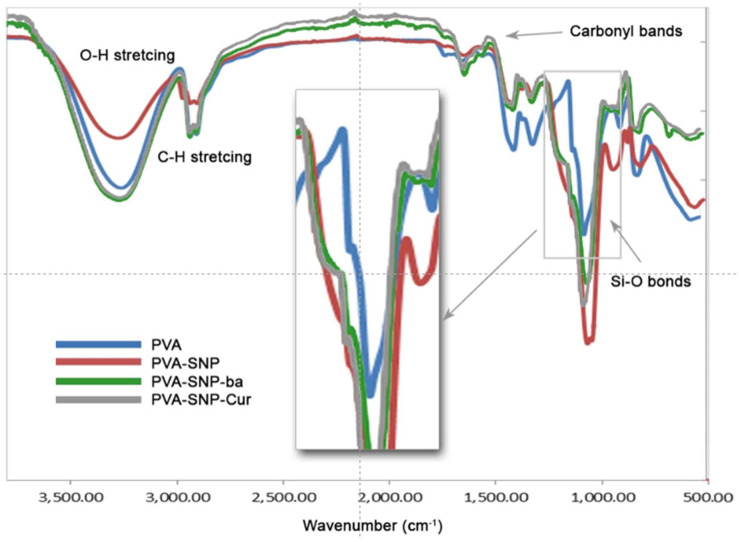
ATR-FTIR spectra of PVA, PVA-SNP, PVA-SNP-cur and PVA-SNP-ba.

**Figure 3 polymers-13-02889-f003:**
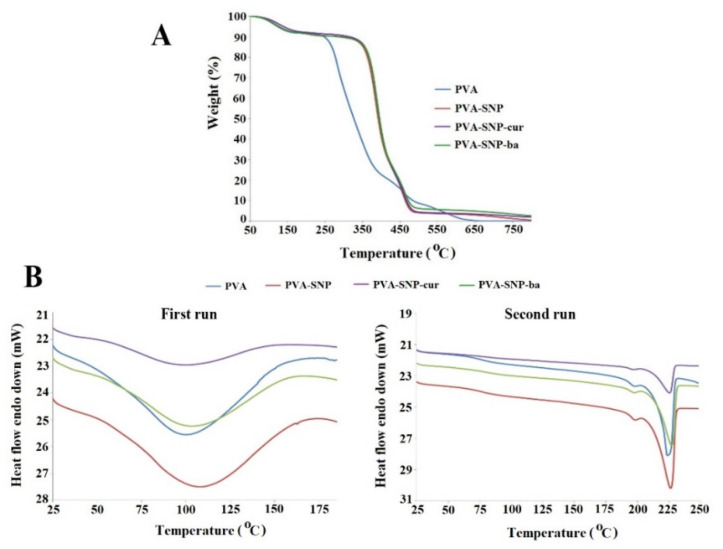
(**A**) TGA spectra of PVA, PVA-SNP, PVA-SNP-cur and PVA-SNP-ba.; (**B**) DSC curves of PVA, PVA-SNP, PVA-SNP-cur, PVA-SNP-ba.

**Figure 4 polymers-13-02889-f004:**
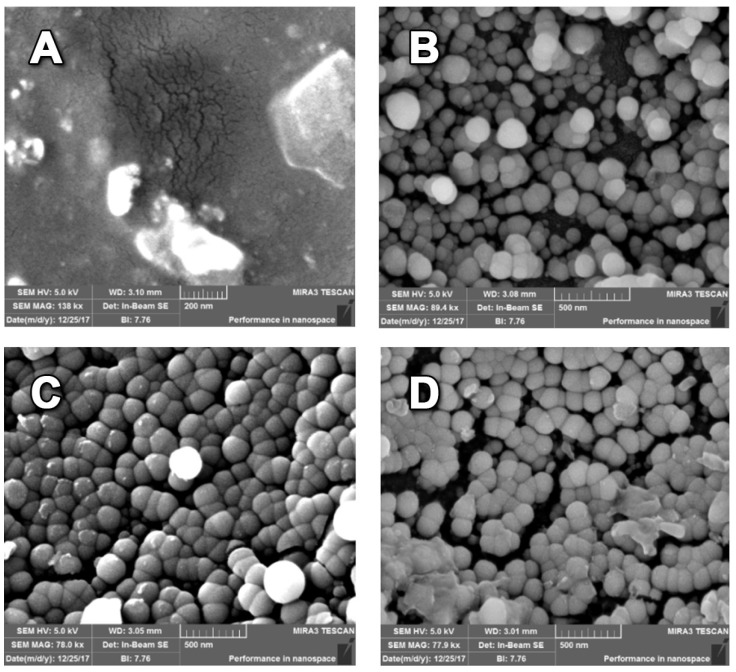
High resolution scanning electron microscopy (HR-SEM) images of (**A**) PVA, (**B**) PVA-SNP, (**C**) PVA-SNP-cur and (**D**) PVA-SNP-ba films.

**Figure 5 polymers-13-02889-f005:**
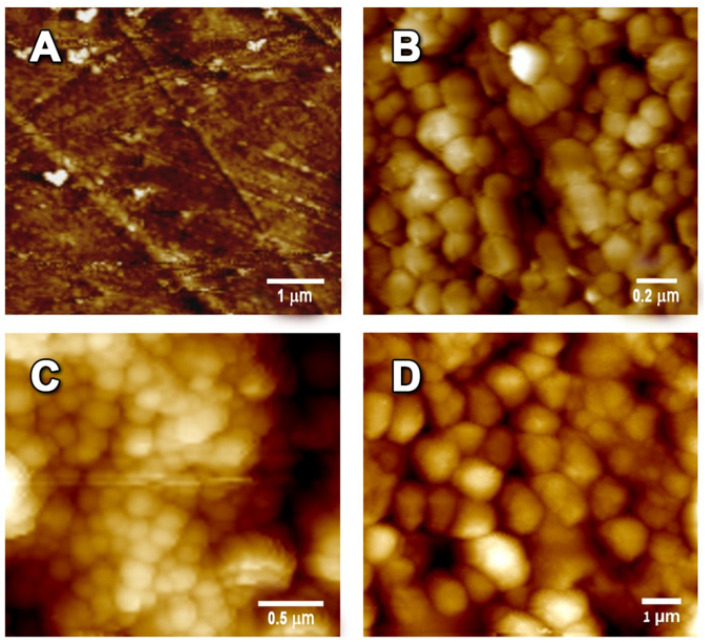
AFM images of (**A**) PVA, (**B**) PVA-SNP, (**C**) PVA-SNP-cur and (**D**) PVA-SNP-ba films measured in tapping mode, with a scan rate of 1.0 Hz at fixed resolution (512 × 512 data points).

**Figure 6 polymers-13-02889-f006:**
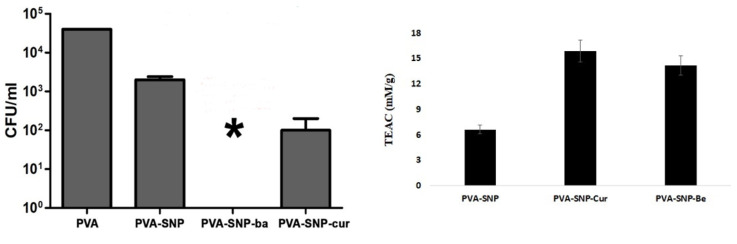
(**Left**) Antifouling activity of PVA, PVA-SNP, PVA-SNP-cur and PVA-SNP-ba *against Listeria innocua*. * Total inhibition of bacterial growth. (**Right**) Antioxidant activity (in Trolox equivalents) in the PVA-grafted films. The TEAC values were calculated using the original unmodified PVA films as a blank.

**Table 1 polymers-13-02889-t001:** Water contact angle, thickness and water vapor permeability (WVP) of the PVA, PVA-SNP, PVA-SNP-cur and PVA-SNP-ba films.

	Thickness (mm)	Contact Angle (degree)	WVP (×10^−10^ g s^−1^ m^–1^ Pa^−1^)
PVA	0.12 ± 0.02	31.71 ± 0.05	0.81 ± 0.07
PVA-SNP	0.21 ± 0.05	86.73 ± 1.31	0.75 ± 0.01
PVA-SNP-cur	0.22 ± 0.05	52.41 ± 0.08	0.46 ± 0.09
PVA-SNP-ba	0.17 ± 0.01	102.16 ± 0.33	0.24 ± 0.04

## Data Availability

The data presented in this study are available on request from the corresponding author.

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
