# Peer review of "In Situ Grafting of Silica Nanoparticle Precursors with Covalently Attached Bioactive Agents to Form PVA-Based Materials for Sustainable Active Packaging"

_polymers, 2021, doi:10.3390/polym13172889_

Round 1

Reviewer 1 Report

In this paper, the authors reported sustainable antibacterial–antioxidant films prepared by in-situ grafting of silica nanoparticle precursors with attached bioactive agents benzoic acid or curcumin on polyvinyl alcohol. The paper fit the aims and scope of Polymers. I would recommend accepting the paper after modifications. I have some comments to the authors.

  1. The development of edible packaging materials (inevitable biodegradable) should be introduced. Some very recent literatures should be mentioned.

doi:10.3390/foods9040449, doi:10.1016/j.jfoodeng.2021.110697, doi:10.1016/j.foodhyd.2018.11.051

  1. Bioactive substances based on natural ingredients capable of extending the food shelf life in a safe manner, especially antibacterial and antioxidant materials, should be used as examples.

doi: 10.1016/j.lwt.2021.111617, doi 10.1021/acs.jafc.0c00945

  1. it is strongly suggested to indicate at the end of the Introduction section the main employed characterisation techniques in order to achieve their purpose.
  2. The detail information of PVA should be given such as; molecular weight, etc.
  3. The method section should be precise, and all the details should be included. For example, measurement wavelength for method of antioxidant activity determination should be provided etc.
  4. Check throughout the text carefully and correct the errors in superscripts and subscripts
  5. ppm (Section 2.3.1) should be changed to a concentration unit in the International System of Units.
  6. The four figures in Figure 2 should be marked separately to clarify what each figure shows. It might be better to mark the characteristic peaks and provide partial enlarged view for FT-IR.
  7. How about the color and transparency of these films? It is recommended to provide photos.
  8. Images under the same measuring scale should be provided for Figure 4.

Author Response

Reviewer 1

Comments and Suggestions for Authors

In this paper, the authors reported sustainable antibacterial–antioxidant films prepared by in-situ grafting of silica nanoparticle precursors with attached bioactive agents benzoic acid or curcumin on polyvinyl alcohol. The paper fit the aims and scope of Polymers. I would recommend accepting the paper after modifications. I have some comments to the authors.

Comment 1: The development of edible packaging materials (inevitable biodegradable) should be introduced. Some very recent literatures should be mentioned.

doi:10.3390/foods9040449, doi:10.1016/j.jfoodeng.2021.110697, doi:10.1016/j.foodhyd.2018.11.051

Response 1: As suggested by the referee, the recent literatures related to development of edible packaging materials were included in the introduction. In the revised manuscript, these new references are numbered 26-28.

Comment 2: Bioactive substances based on natural ingredients capable of extending the food shelf life in a safe manner, especially antibacterial and antioxidant materials, should be used as examples.

doi: 10.1016/j.lwt.2021.111617, doi 10.1021/acs.jafc.0c00945

Response 2: As suggested by the referee, the 2 mentioned references were included in the introduction. In the revised manuscript, these references are numbered16-17.

Comment 3: It is strongly suggested to indicate at the end of the Introduction section the main employed characterization techniques in order to achieve their purpose.

Response 3: As suggested by the referee, detailed explanation of the employed characterization techniques was added at the end of the introduction section.

Comment 4: The detail information of PVA should be given such as; molecular weight, etc.

Response 4: The detailed information of molecular weight (Mw 89,000-98,000), color (white to off-white) and degree of hydrolysis (99+% hydrolyzed) were added in section 2.1

Comment 5: The method section should be precise, and all the details should be included. For example, measurement wavelength for method of antioxidant activity determination should be provided etc.

Response 5: The methods section was revised. The measurement wavelength used to determine the antioxidant activity was added in the methods section and also in Chapter 2.5: "A 734". The subscript error was corrected.

Comment 6: Check throughout the text carefully and correct the errors in superscripts and subscripts

Response 6: All the manuscript was revised and the errors in superscripts and subscripts were corrected.

Comment 7: ppm (Section 2.3.1) should be changed to a concentration unit in the International System of Units.

Response 7:

All NMR measurements are uniformly reported in ppm units. These data represent chemical shift of the measured atom, actually it is the frequency at which the atom undergoes resonance. Despite these data represent frequency, the sifts do not reported as Hertz but using parts ppm (per million) units. The reason is that the measured chemical shifts depend on the strength of magnetic field of a spectrometer. In order to express the same chemical shift values regardless the spectrometer being used, the chemical shifts are automatically converted from Hertz to ppm. It is done by dividing the chemical shift [Hertz] by the frequency rating of the specific spectrometer and multiply it by one million. 

https://chem.libretexts.org/Bookshelves/Organic_Chemistry/Map%3A_Organic_Chemistry_(McMurry)/13%3A_Structure_Determination_-_Nuclear_Magnetic_Resonance_Spectroscopy/13.02%3A_The_Chemical_Shift#:~:text=The%20scale%20is%20commonly%20expressed,absorptions%20occur%20is%20quite%20narrow.&text=For%2013C%20NMR%20almost,the%20C%20atom%20in%20TMS.

Comment 8: The four figures in Figure 2 should be marked separately to clarify what each figure shows. It might be better to mark the characteristic peaks and provide partial enlarged view for FT-IR.

Response 8: As suggested by the referee, Figure 2 was separated to two figures (Figures 2 and 3). The characteristic peaks were marked on the figures. The numbering of the other figures in the whole manuscript where changed accordingly. In the Figure 2 that presents FT-IR spectra, the Si-O bands area was enlarged.

Comment 9: How about the color and transparency of these films? It is recommended to provide photos.

Response 9: The prepared films were colorless and transparent except the PVA-SNP-cur which was yellowish. These details were added to section 3.1 of the revised manuscript.

Comment 10: Images under the same measuring scale should be provided for Figure 4.

Response 10: The measuring scale units were changed according to the referees' comment. In the revised manuscript they are displayed in the same µm units.

Reviewer 2 Report

Dear Editor, dear Authors, Miri Klein et al. submitted a paper on the preparation and characterization of sustainable antibacterial–antioxidant films. The authors used in-situ grafting of stable silica nanoparticle (SNP) modified covalently with antibacterial/antioxidant agents mainly benzoic acid (ba) or curcumin (cur) on a biocompatible polymer polyvinyl alcohol (PVA) by a sol gel method. They have demonstrated that the resulting PVA-SNP, PVA-SNP-cur and PVA-SNP-ba films exhibit antibacterial and antioxidant activities as well as an increased hydrophobicity, resulting in protection from undesired moisture and of the foodborne human pathogen Listeria innocua,, making them a promising PVA-based packaging materials. For my opinion, the authors have performed an experimental work that falls within the scope of Polymers MDPI Journal. Moreover, the results in the manuscript are well supported with experimental evidence. Therefore, I believe that the manuscript can be accepted for publication in Polymers MDPI Journal. Some typing mistakes are stated below should be revised.

Comments

  • Page 3, line 109 “(81μL, 0.326 mmol) and triethylamine (38μL, 0.271 mmol)”add space between the numbers and μL, revise in the whole manuscript.
  • Page 3, line 116 “1H-NMR” use superscript for 1 and same in the line 142 and the rest of the manuscript.
  • Page 3, line 126 “stirred for 30 min at 60 oC, and then 3 mL of aqueous NH4OH (28% w/w in H2O) was” please correct °C and use subscript for 4 and 2 in ammonium hydroxide and water respectively. Same corrections should be done in the lines 151, 159, 160, 292, 294, 295, 302, 306, 309 and the rest of the manuscript. Please revise.
  • Page 3, line 128 “EtOH (x3)” use × instead of x, same in the line 153.
  • Page 3, line 139 “column chromatography on silica gel” The authors here should specify the properties of the used silica gel.
  • Page 4, lines 170-171 “7.26 ppm for CDCl3. All measurements were performed at 22 °C in CDCl3.” Use subscript for 3 in CDCl3
  • Page 4, lines “The films were subjected to 32 scans at 4 cm−1 resolution be-175 tween 500 and 4000 cm−1.” Use superscript for -1 in cm-1, please revise in the whole manuscript e. lines 285, 287, 288, 291 etc.
  • Page 5, line 214 “A is the film area (m2),…” Use superscript for 2 in m2. Similarly for “scanned area (5 x 5 μm2)” in the line 224. Please revise in the whole manuscript.

Sincerely yours,

Author Response

Reviewer 2

Dear Editor, dear Authors, Miri Klein et al. submitted a paper on the preparation and characterization of sustainable antibacterial–antioxidant films. The authors used in-situ grafting of stable silica nanoparticle (SNP) modified covalently with antibacterial/antioxidant agents mainly benzoic acid (ba) or curcumin (cur) on a biocompatible polymer polyvinyl alcohol (PVA) by a sol gel method. They have demonstrated that the resulting PVA-SNP, PVA-SNP-cur and PVA-SNP-ba films exhibit antibacterial and antioxidant activities as well as an increased hydrophobicity, resulting in protection from undesired moisture and of the foodborne human pathogen Listeria innocua, making them a promising PVA-based packaging materials. For my opinion, the authors have performed an experimental work that falls within the scope of Polymers MDPI Journal. Moreover, the results in the manuscript are well supported with experimental evidence. Therefore, I believe that the manuscript can be accepted for publication in Polymers MDPI Journal. Some typing mistakes are stated below should be revised.

Comment 1: Page 3, line 109 “(81μL, 0.326 mmol) and triethylamine (38μL, 0.271 mmol)” add space between the numbers and μL, revise in the whole manuscript.

Response 1: We thank the referee for these corrections. Spaces between the numbers and μL in both cases were added. The whole manuscript was revised.

Comment 2: Page 3, line 116 “1H-NMR” use superscript for 1 and same in the line 142 and the rest of the manuscript.

Response 2: Superscript for 1 in 1H-NMR was done in the mentioned places. In addition, the whole manuscript was revised to correct superscript in 1H-NMR, this correction was also done in section 2.3.1.

Comment 3: Page 3, line 126 “stirred for 30 min at 60 oC, and then 3 mL of aqueous NH4OH (28% w/w in H2O) was” please correct °C and use subscript for 4 and 2 in ammonium hydroxide and water respectively. Same corrections should be done in the lines 151, 159, 160, 292, 294, 295, 302, 306, 309 and the rest of the manuscript. Please revise.

Response 3: The manuscript was revised and all the oC were corrected to °C. In addition, subscript corrections were done for "4" and "2" in "NH4OH (28% w/w in H2O)".

Comment 4: Page 3, line 128 “EtOH (x3)” use × instead of x, same in the line 153.

Response 4: "x" was converted by "×" symbol.

Comment 5: Page 3, line 139 “column chromatography on silica gel” The authors here should specify the properties of the used silica gel.

Response 5: The properties of the used silica gel were specified in section 2.2.2.

Comment 6: Page 4, lines 170-171 “7.26 ppm for CDCl3. All measurements were performed at 22 °C in CDCl3.” Use subscript for 3 in CDCl3

Response 6: Subscript was utilized for 3 in CDCl3 during the whole manuscript.

Comment 7: Page 4, lines “The films were subjected to 32 scans at 4 cm−1 resolution be-175 tween 500 and 4000 cm−1.” Use superscript for -1 in cm-1, please revise in the whole manuscript e. lines 285, 287, 288, 291 etc.

Response 7: Superscript for -1 in cm-1 was used in the whole manuscript. 

Comment 8: Page 5, line 214 “A is the film area (m2),…” Use superscript for 2 in m2. Similarly for “scanned area (5 x 5 μm2)” in the line 224. Please revise in the whole manuscript.

Response 8: The whole manuscript was revised and superscript for "2" was done in "m2"

Reviewer 3 Report

This manuscript portrays the development of novel PVA-based film via in-situ grafting of silica nanoparticles precursors with covalently attached of two different bioactive agents for food packaging. This manuscript is supports with adequate quality analysis and easy to read. However, several concerns need to be address before the manuscript can be accepted for publication. Here are some comments:

Concern:

  1. Line 116 and 142 – Kindly please describe/explain the paragraph so that the reader understands wthe significant why the authors added NMR description.
  2. Line 341 – The authors explain well about the difference in contact angle (CA) result for curcumin (cur) sample with support of curcumin characteristic. But for benzoic acid (ba) sample, no explanation form ba characteristic or properties that can support the enhancement of CA result to the highest value obtained by this study. Hope the author can add.
  3. Line 362 – Please add explanation why the WVP for cur sample have lower value than the PVA-SNP sample. The CA for PVA-SNP is much higher that cur sample, so it should have lower WVP value than cur sample.
  4. Line 372 – more explanation why cur sample could not resist bacterial growth should be added as in the introduction the author already wrote the advantages of curcumin as antibacterial and antiviral. Perhaps due to unoptimized composition?

Author Response

Reviewer 3

Comments and Suggestions for Authors

This manuscript portrays the development of novel PVA-based film via in-situ grafting of silica nanoparticles precursors with covalently attached of two different bioactive agents for food packaging. This manuscript is supports with adequate quality analysis and easy to read. However, several concerns need to be address before the manuscript can be accepted for publication. Here are some comments:

Concern:

Comment 1: Line 116 and 142 – Kindly please describe/explain the paragraph so that the reader understands wthe significant why the authors added NMR description.

Response 1: The paragraph: "1H-NMR measurements were performed verifying the expected chemical structure and purity of the prepared materials. Each chemical shift was related to the appropriate H atom in the prepared precursor molecule. The peaks' split and integration was in accordance to the expected structure."  was added to the revised manuscript.

Comment 2: Line 341 – The authors explain well about the difference in contact angle (CA) result for curcumin (cur) sample with support of curcumin characteristic. But for benzoic acid (ba) sample, no explanation form ba characteristic or properties that can support the enhancement of CA result to the highest value obtained by this study. Hope the author can add.

Response 2: Both benzoic acid and curcumin are hydrophobic molecules, wherefore the increased hydrophobicity of the ba-grafted films is anticipated. On the other hand, the lack of significant hydrophobicity increase in the curcumin-grafted films is more surprising. Clarification of this point was added to the revised manuscript also explaining the differences between curcumin and benzoic acid was added.

 Comment 3: Line 362 – Please add explanation why the WVP for cur sample have lower value than the PVA-SNP sample. The CA for PVA-SNP is much higher that cur sample, so it should have lower WVP value than cur sample.

Response 3: Explanation was added to the revised manuscript. "Interestingly, PVA-SNP films showed higher water vapor permeability than PVA-SNP-cur, despite the fact that PVA-SNP-cur has smaller water contact angle in comparison to PVA-SNP. Probably, in case of PVA-SNP an increase in water contact angle is achieved due to the high surface roughness. However, unlike PVA-SNP-cur or PVA-SNP-ba, PVA-SNP films do not have grafted hydrophobic molecules. Therefore, their ability to inhibit a transport of water molecules is lower than in case of ba or cur-grafted films."

 Comment 4: Line 372 – more explanation why cur sample could not resist bacterial growth should be added as in the introduction the author already wrote the advantages of curcumin as antibacterial and antiviral. Perhaps due to unoptimized composition?

Response 4: The results show that Both PVA-SNP-ba and PVA-SNP-cur have antimicrobial activity. However, PVA-SNP-ba had higher antimicrobial activity then PVA-SNP-cur. It may be because un-optimized composition as the referee suggest or because benzoic acid have higher antimicrobial activity in comparison to curcumin against Listeria innocua. The following explanation was added:" The combination of surface roughness of the silica nanoparticles and the bactericidal properties of the attached active substituent therefore likely conferred antimicrobial properties of the prepared PVA-SNP-ba and PVA-SNP-cur. It can be seen that PVA-SNP-ba had higher antimicrobial activity against Listeria innocua than PVA-SNP-cur. An excellent antimicrobial activity of PVA-SNP-ba correlates with previously reported results, where benzoic acid has been reported to have a strong ability to damage Listeria's phospholipid membrane, perhaps because of its lipophilic features [39]. "